# Optimization of Ultrasound-Assisted Extraction of Some Bioactive Compounds from Tobacco Waste

**DOI:** 10.3390/molecules24081611

**Published:** 2019-04-24

**Authors:** Marija Banožić, Ines Banjari, Martina Jakovljević, Drago Šubarić, Srećko Tomas, Jurislav Babić, Stela Jokić

**Affiliations:** Faculty of Food Technology Osijek, Josip Juraj Strossmayer University of Osijek, Franje Kuhača 20, Osijek 31000, Croatia; marija.banozic@ptfos.hr (M.B.); ines.banjari@ptfos.hr (I.B.); mjakovljevic@ptfos.hr (M.J.); dsubaric@ptfos.hr (D.Š.); srecko.tomas@ptfos.hr (S.T.); jbabic@ptfos.hr (J.B.)

**Keywords:** tobacco waste, ultrasound-assisted extraction, biocompounds, process optimization

## Abstract

This is the first study on ultrasound-assisted extraction (UAE) of bioactive compounds from different types of tobacco industry wastes (scrap, dust, and midrib). The obtained results were compared with starting raw material (tobacco leaves) to see the changes in bioactive compounds during tobacco processing. Results suggested that tobacco waste extracts possess antioxidant activity and considerable amounts of targeted bioactive compounds (phenolics and solanesol). The content of chlorogenic acid varied between 3.64 and 804.2 μg/mL, caffeic acid between 2.34 and 10.8 μg/mL, rutin between 11.56 and 93.7 μg/mL, and solanesol between 294.9 and 598.9 μg/mL for waste and leaf extracts, respectively. There were noticeable differences between bioactive compounds content and antioxidant activity in extracts related to applied UAE conditions and the used type of tobacco waste. Results show that optimal UAE parameters obtained by response surface methodology (RSM) were different for each type of material, so process optimization proved to be necessary. Considering that tobacco waste is mostly discarded or not effectively utilized, the results clearly show that tobacco waste could be used as a potential source of some bioactive compounds.

## 1. Introduction

Tobacco is the most widely cultivated plant in the world which is not consumed as food. Leaves are the most valuable part of tobacco plant and other parts like stem, root, midrib are considered as a waste. Tobacco is part of *Solanaceae* family such as potatoes and tomatoes [1]. Tobacco can be used as a valuable source of bioactive compounds, which could be recovered and used in cosmetics, perfumes and pharmaceuticals [2]. Interest in bioactive compounds extracted from tobacco and tobacco waste increased over the past years [3]. As a result of that growing interest, ultrasound-assisted extraction (UAE) has been widely used for the extraction of bioactive compounds from tobacco leaves (Table 1). However, there is a limited number of publications about the extraction of bioactive compounds from tobacco waste. Moreover, previously mentioned studies have been focused on tobacco waste in general, and not on different types of tobacco waste. Besides already mentioned UAE, other techniques, like microwave extraction and extraction with supercritical fluids, are gaining more attention within the waste management and utilization procedures. UAE has many advantages due to causing smaller structural and molecular changes in material [4]. Additionally, some reported advantages are higher yield of extracted compounds, shorter extraction time, lower extraction temperatures [5], and reducing cost and volume of the used solvent [6].

Additionally, tobacco is the only plant in the world which accumulates larger amounts of solanesol, polyisoprenoid alcohol, which can be used as a medicine and as a coenzyme Q10 or vitamin K2. Coenzyme Q10 is used for the treatment of Alzheimer’s and Parkinson’s disease [23]. In addition, solanesol can be used in the production of new drugs for AIDS-related viruses and cardiovascular diseases. Furthermore, solanesol has been proven to have significant antibacterial, antifungal, antiviral, anticancer, anti-inflammatory, and anti-ulcer activity [24]. Other solanesol derivatives have other bioactive effects, including anti-oxidative and antiproliferative effects [25]. Commercial solanesol is mostly extracted from *Solanaceae* group of plants because its synthesis is difficult [16]. Hu et al. [26] reported solanesol content in tobacco leaves (0.45%), stalk (0.037%), stem (0.0037%), and root (0.0013%), with significant differences between tobacco varieties due to various genetic and environmental factors.

The tobacco industry produces large amounts of waste, and it is important to find ways for its effective utilization. Having in mind significant amounts of phenolic compounds in tobacco, tobacco waste could be used as a valuable source of these compounds for some other industrial applications. After nicotine and solanesol, polyphenols are the most important bioactive compounds of tobacco. Mayor phenolic compounds in tobacco plant are chlorogenic acid (5-caffeoylquinic acid), rutin, caffeic acid and scopoletin [14]. Chen et al. [27] reported 17 polyphenols isolated from tobacco leaves. They have high antioxidant potential, and caffeic acid has stronger antioxidant activity than chlorogenic acid [28]. In addition, they imply on tobacco quality [29] due to their impact on tobacco flavor, the color of tobacco, bitterness, and antioxidant properties [18].

The aim of this study was to (a) provide UAE of from tobacco waste; (b) identify the targeted compounds (phenolics and solanesol) in obtained extracts by using HPLC; (c) compare the results between 3 types of tobacco waste composed by scrap, midrib and dust, generated during tobacco processing, and leaves as starting material; and (d) define optimum parameters for the UAE to obtain maximal amounts of selected bioactive compounds in extracts. 

## 2. Results and Discussion

The UAE was performed in order to research possibilities for extracting different bioactive compounds from tobacco waste. Extraction parameters were defined by Box-Behnken (BBD) experimental design (Table 2 and Table 3). Two different extraction procedures were used, one for extraction of phenolic compounds, and the other for extraction of solanesol. 

For extraction of phenolic compounds, four (4) parameters of extraction were changed, namely, temperature in the range of 30–70 °C, time from 15–45 min, solvent-solid ratio from 10–30 mL/g and ethanol-water ratio (*v*/*v*) from 40–80%. Ethanol was chosen to be extraction solvent because aqueous dilution of ethanol is the most frequently used solvent for the extraction of phenolic compounds from plant and plant related materials. Those type of dilutions can dissolve a wide range of phenolic compound and they also represent a cheap green solvent, acceptable for human consumption [30,31,32]. 

During the extraction of solanesol, the solvent-solid ratio and ethanol-water ratio (*v*/*v*) were constant according to the research of other authors [33] where the hexane-ethanol ratio (1/3) and solid-solvent ratio 1:40 gave the highest yield of extracted solanesol. During the experiment, temperature varied between 33.8 and 76.2 °C and time varied between 8.8 min and 51.2 min, as shown in Table 3. It is well known that UAE reduce extraction time and produce the higher yield of extracted phenolic compounds due causing structural changes and cell disruptions such that the phenolic compound can be easily be dissolved in solvent [8]. Often mentioned disadvantages of this method are related with scaling up. There are a number of different ultrasonic devices on the market. With respect to extraction conditions, type of extractor play a very important role in extraction efficiency. In comparison with flow-reactors, batch reactors are found less effective in extraction yield, solvent consumption, and time. There are limited numbers of reactors which are appropriate for the industrial scale [34,35,36]. Alexandru et al. [36] found that only flow reactors are suitable for scaling up. Therefore, future industrial application of ultrasound-assisted extraction probably would be based on this technology.

### 2.1. Detected Bioactive Compounds in Tobacco Leaf and Waste Extracts

Targeted phenolic compounds and solanesol were identified by HPLC. HPLC is the most common analytical technique for the identification and characterization of phenolic compounds in plant materials. A chromatographic profile of 80% ethanol extract, obtained under the following conditions of 30 min, 30 °C and using 30 mL/g, is presented in Figure 1. As can be seen from Figure 1, six (6) peaks were observed at 210 nm, and three (3) of them were identified. The largest peak was identified as chlorogenic acid (peak 1). The second peak was identified as caffeic acid and the third one was identified as rutin. Targeted compounds were identified by retention times and a UV spectra by comparing them with commercial standards, as well as comparing them with data available in the literature [14,22,37,38]. Moreover, a chromatogram of the obtained extract shows two peaks near to the chlorogenic acid. 

Chlorogenic acid (5-caffeoylquinic acid) is a part of a family of chlorogenic acids, which also include cryptochlorogenic acid (4-CQA), neochlorogenic acid (3-caffeoylquinic acid, 3-CQA), and 1-caffeoylquinic acid (1-CQA). Products of their hydrolysis are caffeic acid [22]. The highest chlorogenic acid content was achieved at following conditions of temperature 50 °C, extraction time 30 min, solvent-solid ratio of 10 mL/g, and ethanol-water ratio of 60% for leaves extracts, and at same conditions for scrap and dust extracts as well. The content of chlorogenic acid varied between 44.3 and 2720 μg/mL in leaves extracts, and between 3.64 and 804.2 μg/mL for waste extracts, which depended on extraction condition but also on the type of waste. Similar results were reported by Wang et al. [39], where the solvent-solid ratio had the highest influence on chlorogenic acid, which clearly indicates that decrease of solvent improves of chlorogenic acid content.

Low content of caffeic acid in all extracts, including tobacco leaves (3.6–31.5 μg/mL) and waste extracts (2.34–10.8 μg/mL), could also be explained by the degradation of caffeic acid under the ultrasound treatment. It has been shown that caffeic acid content decreases during UAE. Despite their chemical structures, caffeic acid and their analogs are very thermolabile [38]. Also, they reported lower contents of caffeic acid ranging from 0.44–2.07 μg/mL in tobacco waste, and Liu et al. [37] reported 5.6 μg/mL in tobacco water extracts. Xie et al. [14] reported higher contents of caffeic acid in tobacco leaves (0.007–0.025%). 

The content of rutin in tobacco leaves varied between 77.6 μg/mL and 427 μg/mL, similar to the results by Fathiazad et al. [40], where the rutin content was 0.5%. The highest rutin content in this study showed leaf extract, extracted under the condition of temperature 50 °C, extraction time 45 min, solvent-solid ratio 50 mL/g, and ethanol-water ratio of 40%. Considering waste extracts, rutin content varied between 14.29 and 93.7 μg/mL for scrap extracts, from 12.77 and 58.20 μg/mL for dust extracts, and for midrib extract, they were very low up to 11.56 μg/mL, and for some midrib extracts it was under the limit of detection. Those results are slightly lower than those reported by Fathiazad et al. [40], where rutin content in tobacco waste extracts was 0.6%.

Various protocols have been reported for the extraction of solanesol [7,33,41,42], and all of them include a saponification step, since solanesol is mostly bounded in the form of an ester. Therefore, saponification has been carried out in all previously mentioned studies. Solanesol content in obtained tobacco leaves extracts in this study ranged from 294.9–598.9 μg/mL (0.8–2.3%), which is similar to results published by Chen et al. [7], where they reported 0.30–3.0% of solanesol content, but slightly higher than in the study by Hu et al. [26], reporting 0.45%, or Zhao et al. [43], reporting 0.57–0.64% of solanesol in the tobacco leaves. Figure 2 shows a chromatogram of the obtained scrap extract (Run 1), where peak 4 was identified as solanesol. 

Considering tobacco waste as a source of solanesol, it can be noticed that solanesol content is significantly lower than in leaves extracts. There is no available data on the single fraction of tobacco waste to compare our results with. Similar results were reported by Atla et al. [44], where solanesol content in tobacco waste was 0.05%. It can be noticed that there were no significant differences between solanesol contents in scrap (114.4–162.5 μg/mL) and dust (81.7–182.8 μg/mL), but in midrib extracts, solanesol content was significantly lower (50.3–62.7 μg/mL). Applied ultrasound conditions of temperature and time did not significantly influence the solanesol content. However, the obtained results show that properly conducted pre-treatment with the saponification step, well-chosen solvent, and the solvent-solid ratio are key steps in the maximum extraction yield of solanesol, and time and temperature have positive influences (but not statistically significant).

### 2.2. Antioxidant Activity of Tobacco Leaf and Waste Extracts 

The DPPH assay is one of the most widely used methods for the determination of antioxidant activity. It measures the ability of antioxidant compounds to reduce free radicals. Degrees of inhibition of DPPH for waste samples varied from 0.71–37.24%, and for leaves extracts from 13.94–95.02%. These results are similar to research by Cvetanovska et al. [19], where the methanolic leaf extract showed radical scavenging activity of 78 ± 2.20%. The highest degree inhibition of DPPH (37.24 ± 1.19%) from waste extracts showed scrap extract extracted under the following extraction conditions: Temperature 50 °C, time 30 min, solvent-solid ratio 10 mL/g, and ethanol-water ratio 40%. From the leaves extracts, the highest degree inhibition of DPPH (95.02% ± 0.13) showed the extract obtained under following conditions: Temperature 50 °C, time of extraction 30 min, solvent-solid ratio 50 mL/g, and ethanol-water ratio 40%. It can be noticed that the ultrasound conditions for both were the same for temperature, time, and ethanol-water ratio conditions. The antioxidant activity in the obtained extracts was the highest in leaves extracts followed, by scrap extracts. In the present study it is likely that, apart from the process UAE condition, the type of tobacco waste significantly influenced antioxidant activity. 

### 2.3. Statistical Analysis

Differences between tobacco leaves as the raw, starting material, and tobacco wastes in general will be discussed below. All waste extracts were combined for the purpose of more thorough and precise statistical analysis. Observed parameters related to tobacco waste and leaves are shown in Table 4. Ranges give insight into the wide distribution of observed values, especially if the values between waste and leaves were considered.

No significant correlation was found between antioxidant activity (observed as DPPH value) and bioactive compounds in leaves extracts, but did reveal significant correlations for waste extracts (Table 5). Content of chlorogenic acid (r = 0.537) and rutin (r = 0.407) significantly influences the antioxidant activity of the waste extracts.

Moreover, leaves contain other unidentified compounds, behind chlorogenic acid and rutin, which might contribute to the overall antioxidant properties. Popova et al. [21] reported significant amounts of other phenolic compounds in tobacco extracts, like salicylic, proto–catechuic, and p–Coumaric acid. Moreover, Docheva et al. [9] reported that with an increase of flavonoid content in tobacco and tobacco waste extracts, the antioxidant activity of extracts also increased. However, tobacco leaves contain carotenoids such as lutein, P–carotene, neoxanthin, and violaxanthin, which also might affect antioxidant activity [45]. Thus, more studies should analyze the role of those compounds on the antioxidant activity of tobacco.

Correlations between selected parameters and ethanol-water ratio and solvent-solid ratio are shown in Table 6. For leaves and waste extracts, the ethanol-water ratio and solvent-solid ratio negatively influence the content of chlorogenic acid. For leaves extracts, the same negative effect is visible for caffeic acid (though not significant), while a positive trend can be noted for waste extracts. The solvent-solid ratio also shows a positive effect on the content of rutin in both types of extracts. These results enable fine optimization of the extraction process, i.e., it is impossible to use one, universal extraction method for all phenolic compounds. Therefore, the extraction method should be optimized according to the targeted bioactive compound. However, if chlorogenic acid would be the targeted compound, then other phenolic compounds would be undesirable in extracts, and a selective extraction technique would be more appropriate. 

Process parameters were also observed from the aspect of solanesol content. While for waste extracts no significant correlation was found between the content of solanesol and processing parameters (temperature and time), both process parameters were found to significantly positively influence the content of solanesol in leaves extracts (r = 0.611 for temperature and r = 0.593 for time, Speaman’s Correlation Coefficients significant at *p* < 0.05). In other research it has been reported that prolonged time increased solanesol content. On the other hand, prolonged time also increased theamount of used hexane, which probably had more influence on solanesol content than prolonged time [20].

Additionally, we looked at the differences in the content of bioactive compounds depending on the process parameters of leaves and waste extracts. Processing parameters were: Time: 15 min, 30 min and 45 min; temperature 30 °C, 50 °C and 70 °C; solvent-solid ratio 10 mL/g, 30 mL/g, and 50 mL/g; ethanol-water ratio 40%, 60%, and 80%. In both types of extracts, changes in time and temperature gave no significant differences in bioactive compounds. Prolonged time and higher temperatures did not result in increases of targeted bioactive compounds.

The ethanol-water ratio showed a significant impact only on the content of chlorogenic acid in both types of samples. For leaves samples, the highest content of chlorogenic acid was gained with 40% ethanol-water ratio (142.83 ± 71.78 μg/mL; *p* = 0.043; *t*-test for independent samples), while for waste samples, the highest content was gained with 60% ethanol-water ratio (*p* = 0.008; Kruskal–Wallis ANOVA test).

The highest chlorogenic acid content was extracted at low solvent-solid ratio (10mL/g) while the highest content of the extracted rutin, was gained at higher solvent-solid ratio (50mL/g) as been shown in Table 7.

### 2.4. Optimization of Extraction Conditions

In order to get higher amounts of desired targeted compounds, chlorogenic acid, and rutin, extraction parameters were optimized using RSM. These compounds were selected because during the test it has been shown that the extraction parameters significantly affect their amounts. Optimal conditions for the extraction of chlorogenic acid and rutin from tobacco waste and leaves depended on temperature (°C), time (min), solvent-solid ratio (mL/g), and ethanol water ratio (%), and are presented in Table 8. Optimal temperatures for chlorogenic acid were in ranges from 30.14–53.59 °C, and for rutin from 37.78–69.72 °C. Optimal solvent-solid ratios were low for chlorogenic acid (in the range 10–11 mg/mL) and very high for rutin extraction (in the range from 49.06–49.88) for all types of wastes and leaves, as confirmed by the statistical analysis (Table 7). Extraction of chlorogenic acid takes less time (in the range from 15.19–38.31 min) than the extraction of rutin (in the range from 26.23–43.02 min). For the extraction of chlorogenic acid, a lower Ethanol-water ratio in the solvent is required (in the range from 40–55.43%), while for the rutin, a higher ethanol-water ratio optimal (in a range from 58.43–69.72%).

Our present results clearly indicated that there is a substantial difference in the UAE conditions for different types of tobacco wastes. In addition, differences in the amounts of desired components are also significant. Comparing all 3 types of waste, the scrap proved to be best for the extraction of rutin and chlorogenic acid, followed by dust, while midrib had the lowest content of desired compounds.

## 3. Materials and Methods 

Tobacco leaves and wastes (midrib, dust and scraps) were obtained from a tobacco processing factory “Fabrika duhana Sarajevo” (Sarajevo, Bosnia and Herzegovina) in 2018. Detailed descriptions of used materials are given in our previously published paper [46]. All samples were kept at ambient temperature in a dark and dry place. Tobacco leaves and wastes were pulverized before extraction. The standard of solanesol (purity 90) was purchased from Sigma-Aldrich (Taufkirchen, Germany) and chlorogenic acid (purity 96.63%, caffeic acid (purity 97.7%) and rutin (94.56%) were purchased from Dr. Ehrenstorfer GmbH (Augsburk Germany). All used solvents were of HPLC grade and they were purchased from J.T.Baker (New Jersey, PA, USA). Other used chemicals were of analytical grade. 

### 3.1. Ultrasound-Assisted Extraction (UAE) of Phenols

For the extraction of phenols, the process was performed at 3 different temperatures (30, 50, and 70 °C), for different extraction times (15, 30, and 45 min), the solvent-solid ratio (10, 30, 50 mL/g), and the ethanol-water ratio (60:40, 40:60, 20:80). Extraction was performed in an ultrasound-bath Elma, Elmasonic P 70 H, with a frequency of 37 kHz and a power of 50 W. Afterwards, the obtained extracts were filtered through filter paper and stored at 4 °C. According to the Box-Behnken design, 29 experiments for every type of sample (leaves and 3 type of wastes) were performed in a total of 116 experiments.

### 3.2. Ultrasound-Assisted Extraction of Solanesol

In tobacco leaves, solanesol can be found in two forms: In a free state and as a fatty acid ester. Saponification is carried out to improve the extraction of solanesol during the adding of NaOH. It has been shown in a published paper by Zhou et al. [33] that a solid-solvent ratio (hexane:ethanol) of 1:3 with 0.05 mol/L of NaOH gave the best results for the extracting of solanesol. Experimental design was performed with 13 experiments and two studied variables (extraction time and temperature) with 5 replicates. One g of powdered material was added into 40 mL solvent (ethanol:hexane, 3:1 *v*/*v*) following to procedure described in previously mentioned research. Extraction were performed at the same system as extraction of phenols with different numbers of Experiment (13). According to Box-Behnken design, 13 experiments were performed for every type of tobacco sample, in a total 52 experiments. After extraction, extracts were centrifuged 5 min at 4250 g and stored at 4 °C until HPLC analysis.

### 3.3. HPLC Analysis

The content of polyphenols and solanesol was determined using HPLC (Agilent technologies 1260 Infinity II, Santa Clara, CA, USA). The polyphenol content was determined with modified method described in Wang et al. [22], under the following conditions: Mobile phase: 0.1% phosphoric acid:methanol (55:45 *v*/*v*), injection volume 20µl and flow rate 0.5 mL/min, while time of analysis was 10 min. Analysis was monitored at 210 nm on DAD detector. 

Solanesol was determined using modified method described in Zhou et al. [33], under the following conditions: Mobile phase: Isopropanol:methanol (40:60 *v*/*v*), injection volume 20 µL and flow rate 0.5 mL/min, time was 10 min. Analysis was monitored at 214 nm on DAD detector (Santa Clara, CA, USA).

The used column was a Zorbax Eclipse Plus C18 (particle size 10 mm × 4.6 mm, 5 µm, Santa Clara, CA, USA) and chromatography was performed at room temperature (22 °C). Standard stock solutions for chlorogenic and caffeic acid, rutin, and solanesol was prepared in water for polyphenols and methanol for solanesol and calibration was obtained at eight concentrations (concentration ranges of 20.0, 30.00, 50.0, 75.0, 100.0, 150.00, 200.0, mg/L). Linearity of the calibration curve was confirmed by R^2^ = 0.99940 for chlorogenic acid, R^2^ = 0.99911 for caffeic acid, and R^2^ = 0.99890 for rutin. Chlorogenic acid limit of detection (LOD) was 0.00036 mg/L, limit of quantification (LOQ) was 0.0012 mg/L and compound retention time was 2.564 min. Caffeic acid limit of detection (LOD) was 0.0011 mg/L, limit of quantification (LOQ) was 0.0037 mg/L and compound retention time was 3.430 min. Rutin limit of detection (LOD) was 0.0020 mg/L, limit of quantification (LOQ) was 0.0068 mg/L and compound retention time was 6.704 min. Limit of detection for solanesol (LOD) was 0.47282 mg/L, limit of quantification (LOQ) was 1.57605 mg/L, and compound retention time was 7.567 min.

### 3.4. Antioxidant Activity of Obtained Extracts

The DPPH method was performed to determine the antioxidant activity of the extracts. For evaluating the free radical scavenging activity we used DPPH. The methanol solution of DPPH was prepared daily and kept in a dark place. Extracts (1.2 mL) with added DPPH (0.5 mL) were kept in dark place for 30 min. Absorbance was taken after 30 min at 517 nm using a spectrophotometer [47]. All measurements were done in triplicate, compared with control blank and calculated using the following Equation (1):(1)$$\%\;\mathrm{DPPH}\;\mathrm{activity}=\frac{\left(A_{DPPH}+A_{b}\right)-A_{s}}{A_{DPPH}}\times 100$$

### 3.5. Statistical Analysis

Statistical analysis of the collected measured data was performed with software tools Statistica 13.3 (TIBCO Software Inc., Palo Alto, CA, USA) at a significance level *p* = 0.05. The normality of data distribution was tested by the nonparametric Kolmogorov-Smirnov test for the comparison of medians and arithmetic mean, and histograms plotting. Due to variable distribution, both parametric and nonparametric statistical tests were combined. Numerical variables are shown as arithmetic means and standard deviations, along with the minimum and maximum values. Besides descriptive statistical analysis and correlation calculations, a comparison of independent variables and the ANOVA test were used.

### 3.6. Optimization of the Extraction Process

Optimization of UAE was performed using response surface methodology (RSM) and BBD based on different extraction parameters (temperature, time, solid/solvent and ethanol-water ratio as independent variables and their influence on polyphenol and solanesol content and antioxidant activity as responses. Optimization was conducted in order to get higher amounts of targeted phenolic compounds and solanesol using a desirability function.

## 4. Conclusions

Taking into account the numerous advantages of UAE, such as reduced extraction time, lower temperature, and acceptable solvent consumption, this method should be considered as an effective alternative method for the extraction of solanesol and phenolic compounds from tobacco wastes. To the best of our knowledge, this is the first study which looked into extraction and composition of bioactive compounds in different types of tobacco wastes using UAE. All bioactive compounds studied, which were found in tobacco leaves, are also present in all fractions of tobacco waste, but in lower concentrations. Consequently, the content and antioxidant activity of these bioactive compounds were higher in leaves than in waste extracts. Also, considering waste samples, scrap and dust samples are more appropriate for the extraction of bioactive compounds than midrib. Tobacco waste could be used as sources of bioactive compounds, which could be implemented in some products, such as pharmaceutical or cosmetics products. However, more studies are encouraged to elucidate the complexity and effects of tobacco waste extracts. Further work should be directed toward isolation and characterization of other phenolic compounds in tobacco waste, their effects on the antioxidant activity, and also a comparison with other green extraction methods, such as microwave-assisted extraction, supercritical fluid extraction, and subcritical water extraction. 

## Figures and Tables

**Figure 1 molecules-24-01611-f001:**
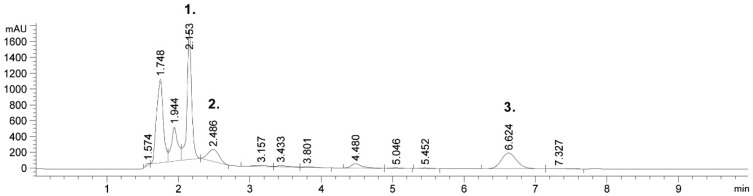
Chromatogram of phenolic compounds in scrap extract: Run 9.

**Figure 2 molecules-24-01611-f002:**
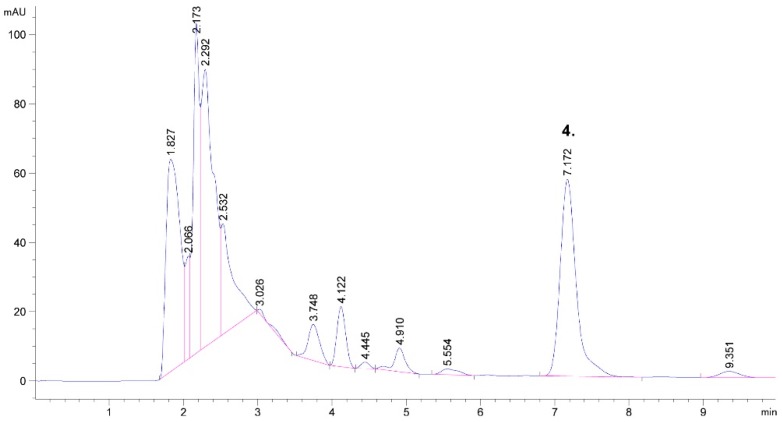
The chromatogram of solanesol in tobacco waste (scrap): Run 1.

**Table 1 molecules-24-01611-t001:** Ultrasound-assisted extraction (UAE) of bioactive compounds from tobacco and tobacco-related material: An overview from available literature.

Plant Part	Process Parameters of UAE					Extracted Compounds	Reference
	Temperature (°C)	Time (min)	Solvent	Solid-Solvent Ratio	Frequency/Power		
Leaves and residues	-	60	Ethanol	1:10	-	Solanesol	[7]
Residues	25	15	Acetone:water (1:2 *v*/*v*)	1:9	100 W	Polyphenols	[8]
Leaves and waste	40	30	Ethyl acetate:methanol (1:1 *v*/*v*)	1:30	-	Phenolic acids Flavonoids	[9]
Leaves	25 and 40	20	Acetone Methanol	1:10	40 Hz	Phenols and flavonoids	[10]
Leaves	40	30	Ethyl acetate:methanol (1:1 *v*/*v*)	1:50	-	Flavonoids	[11]
Leaves	35	30	Anhydrous methanol with 0.5% ascorbic acid	1:10	-	Polyphenols	[12]
Leaves	30–50	5–30	Water + 10–15 drops of ethanol	20:256	24 kHz	Chlorogenic acid	[13]
Leaves	-	30	Methanol:water (70:30 *v*/*v*)	1:160	50 Hz	Polyphenols	[14]
Different part of tobacco plant	45	15	Ethanol:water (85:15 *v*/*v*)	1:15	40 Hz	Solanesol	[15]
Leaves	-	30	Acetone	1:30	-	Fatty alcohols Phytosterols Solanesol	[16]
Leaves	70	180	Methanol:water (70:30 *v*/*v*)	-	-	Polyphenols	[17]
Cigarette tobacco	-	30	-	1:45	200 W	Rutin	[18]
Lyophilized leaves	-	15	Methanol:water (80:20 *v*/*v*)	-		Polyphenols	[19]
Fresh leaves	60	240	Ethanol: hexane (125:75 *v*/*v*)	10:200	47 ± 3 kHz	Solanesol	[20]
Leaves	60	3 × 20	Methanol:water (60:40 *v*/*v*)	1:50	-	Phenolic acids	[21]
Waste material	-	120	Methanol:water (70:30 *v*/*v*)	150:500	-	Chlorogenic acids and rutin	[22]

**Table 2 molecules-24-01611-t002:** Detected phenolic compounds in tobacco extracts and its antioxidant activity.

RUN	UAE Conditions				Chlorogenic Acid (ug/mL)				Caffeic Acid (ug/mL)				Rutin (ug/mL)				DPPH (%)			
	Temperature (°C)	Time (min)	Solvent-Solid ratio (mL/g)	Ethanol-Water ratio (%)	Leaves	Scrap	Dust	Midrib	Leaves	Scrap	Dust	Midrib	Leaves	Scrap	Dust	Midrib	Leaves	Scrap	Dust	Midrib
**1.**	70	30	30	80	143.0	25.0	23.01	11.35	10.6	8.0	-	-	119	37.8	25.31	10.38	56.28	3.40	10.38	7.61
**2.**	50	30	30	60	251.8	115.4	103.8	19.40	11.1	3.4	-	-	126	38.7	29.67	20.76	79.69	25.1	20.76	17.51
**3.**	50	30	30	60	353.4	111.9	106.8	70.75	8.7	7.1	-	-	89.6	38.6	28.78	22.81	80.63	25.1	22.81	14.95
**4.**	70	45	30	60	297.0	116.3	115.1	20.69	12.8	5.7	-	-	134	36.0	32.57	26.87	91.01	29.1	26.87	10.00
**5.**	30	30	30	40	434.3	103.6	96.95	59.93	9.9	6.2	-	-	85.2	29.9	18.68	25.23	87.26	26.9	25.23	17.68
**6.**	50	15	10	60	2221	556.4	429	141	31.5	-	-	-	281	81.9	43.10	19.48	88.95	34.5	19.48	13.55
**7.**	70	30	50	60	74.9	15.5	15.24	-	4.6	4.9	3.44	-	116.1	47.7	34.68	16.09	85.01	21.4	16.09	18.36
**8.**	50	30	30	60	357.4	112.1	100.8	24.12	8.8	5.0	-	-	99.5	35.8	29.83	21.33	76.85	25.5	21.33	18.33
**9.**	30	30	30	80	69.7	21.7	19.77	8.28	8.7	-	-	-	85.2	14.9	12.77	0.71	13.94	3.36	0.71	6.14
**10.**	50	30	30	60	281.0	112.3	101.2	17.89	10.9	6.4	-	-	121.4	43.0	23.03	23.95	75.23	24.2	23.95	15.39
**11.**	50	30	10	40	2720.0	804.2	508.4	128.7	31.5	-	-	-	311.0	68.8	41.50	28.49	94.94	37.2	28.49	24.74
**12.**	50	30	50	80	44.3	26.3	8.04	-	-	3.3	5.20	3.32	123.5	34.4	33.78	14.85	78.83	18.3	14.85	7.41
**13.**	50	30	50	40	102.8	8.3	22.30	11.86	8.9	7.4	-	5.70	99.6	43.6	33.40	26.54	95.02	20.8	26.54	18.43
**14.**	50	45	50	60	75.0	10.9	13.60	5.06	6.5	10.8	5.48	3.68	138.1	42.7	34.76	23.11	80.37	16.99	23.11	13.41
**15.**	30	15	30	60	193.0	121.7	105.9	41.15	6.4	-	-	-	91.5	33.4	19.60	23.98	94.19	16.18	23.98	9.11
**16.**	50	15	50	60	76.3	16.2	11.84	-	4.8	3.0	2.34	3.20	119.9	42.4	30.12	19.42	78.42	28.74	19.42	11.47
**17.**	50	45	30	80	165.0	99.7	85.40	8.78	12.1	6.9	-	7.89	171.6	43.7	32.01	18.58	93.86	14.89	18.58	17.75
**18.**	50	45	30	40	463.8	106.6	50.56	19.73	9.1	6.4	-	-	88.8	28.9	23.00	27.28	94.01	32.80	27.28	19.86
**19.**	50	45	10	60	2008	557.7	458.0	164.5	25.5	-	-	-	299.0	48.5	58.20	24.89	94.64	17.47	24.89	21.19
**20.**	30	30	50	60	54.9	17.8	13.66	3.64	3.6	3.2	3.66	-	116.7	47.6	33.80	20.62	93.44	26.45	20.62	11.30
**21.**	50	15	30	80	77.3	22.9	23.20	8.25	11.4	-	-	-	77.6	22.5	25.64	11.52	26.08	10.01	11.52	5.70
**22.**	70	30	30	40	393.2	113.9	49.43	22.90	10.7	7.8	-	-	96.0	28.6	20.26	25.53	74.56	33.32	25.53	17.27
**23.**	70	30	10	60	1944	605.0	462.7	137.1	16.9	-	-	-	327.9	65.1	42.70	20.66	77.56	14.44	20.66	14.44
**24.**	50	30	10	80	1713	424.8	290.3	96.90	24.9	-	-	-	266.5	54.0	41.40	13.37	49.76	13.52	13.37	10.68
**25.**	30	45	30	60	313.3	124.2	103.1	67.78	8.4	6.2	-	-	93.7	30.1	24.92	19.05	78.16	14.33	19.05	12.83
**26.**	50	15	30	40	469.2	106.3	46.50	19.57	9.5	6.6	-	-	82.4	24.9	14.65	24.69	74.48	19.95	24.69	16.59
**27.**	50	30	30	60	303.1	125.8	102.5	21.88	9.6	8.4	-	-	102.7	36.0	26.40	22.07	75.95	26.41	22.07	15.77
**28.**	30	30	10	60	2318.	653.5	397.4	61.60	30.7	-	-	-	427.6	93.7	32.80	18.68	87.19	24.34	18.68	11.88
**29.**	70	15	30	60	228.4	116.1	105.2	44.35	11.4	7.4	-	-	128.5	47.8	26.27	20.89	69.99	26.93	20.89	17.44

**Table 3 molecules-24-01611-t003:** Solanesol content in tobacco extracts.

RUN	UAE Conditions		Solanesol (μg/mL)			
	Temperature (°C)	Time (min)	Leaves	Scraps	Dust	Midrib
**1.**	70	45	556	162.5	148	61.2
**2.**	55	51.2	598.9	148.2	137.7	54.2
**3.**	55	30	485.1	148.4	128.4	53.9
**4.**	55	30	406.2	143.2	125.9	53.8
**5.**	55	30	456.1	135.0	125.8	52.0
**6.**	76.2	30	598.7	129.4	182.8	56.7
**7.**	55	8.8	416.4	114.4	106.4	57.5
**8.**	33.8	30	294.9	127.3	109.4	51.7
**9.**	70	15	457.4	134.5	81.7	55.4
**10.**	40	15	398.7	123.2	96.5	50.3
**11.**	40	45	473.7	136.3	132.7	54.3
**12.**	55	30	539.5	133.5	122.0	62.7
**13.**	55	30	446.2	129.8	92.8	55.5

**Table 4 molecules-24-01611-t004:** Determined bioactive compounds and antioxidant activity in leaves and waste extracts.

Extracted Compound	Leaves Extracts			Waste Extracts			*p*
	Mean ± SD	Min	Max	Mean ± SD	Min	Max	
Chlorogenic acid (μg/mL)	102.05 ± 70.12	21.13	272.09	21.94 ± 17.31	1.82	80.42	<0.001
Caffeic acid (μg/mL)	2.82 ± 0.81	0.00	4.43	2.30 ± 1.14	1.04	6.66	0.048
Rutin (μg/mL)	37.65 ± 13.18	23.52	69.06	8.74 ± 5.67	1.28	23.86	<0.001
Solanesol (ng/μg)	28.63 ± 17.87	29.49	59.89	8.24 ± 3.65	5.03	18.28	<0.001 *
DPPH	77.46 ± 19.41	13.94	95.02	18.86 ± 7.28	0.71	37.24	<0.001 *

* Mann–Whitney U test.

**Table 5 molecules-24-01611-t005:** Correlations between parameters of antioxidative activity in waste extracts.

Extracted Compound	Chlorogenic Acid	Caffeic Acid	Rutin	Solanesol	DPPH
Chlorogenic acid	1.000				
Caffeic acid	−0.282	1.000			
Rutin	0.088	−0.045	1.000		
Solanesol	0.059	−0.079	0.179	1.000	
DPPH	0.537 **	−0.220	0.407 **	0.003	1.000

Spearman’s Correlation Coeficients; ** significant at *p* < 0.01.

**Table 6 molecules-24-01611-t006:** Correlations between ethanol-water ratios and solvent-solid ratios and bioactive compounds.

Extracted Compound	Leaves Extracts *		Waste Extracts **	
	Solvent-Solid Ratio (mL/g)	Ethanol-Water Ratio (%)	Solvent-Solid Ratio (mL/g)	Ethanol-Water Ratio (%)
**Chlorogenic acid** (μg/mL)	−0.839 ^§^	−0.409 ^¥^	−0.556 ^§^	−0.207
**Caffeic acid** (μg/mL)	−0.127	−0.268	0.058	0.035
**Rutin** (μg/mL)	0.685 ^§^	0.158	0.425 ^§^	−0.068

* Pearson’s Correlation Coefficients; ** Spearman’s Correlation Coefficients ^¥^ significant at *p* < 0.05; ^§^ significant at *p* < 0.01.

**Table 7 molecules-24-01611-t007:** The difference in the contents of bioactive compounds depending on the amount of solvent used for the extraction.

Sample	Solvent Content	Chlorogenic Acid (μg/mL)	Caffeic Acid (μg/mL)	Rutin (μg/mL)	*p*
Leaves	10 mL/g	215.45 ± 35.00	2.68 ± 0.57	31.89 ± 5.74	<0.001 ^1^ <0.001 ^2^
	30 mL/g	85.45 ± 37.71	3.03 ± 0.48	31.97 ± 7.48	
	50 mL/g	35.68 ± 10.11	2.37 ± 1.48	59.49 ± 6.23	
Waste	10 mL/g	38.21 ± 21.93	0.00	5.32 ± 2.02	<0.001 ^3^ 0.001 ^4^
	30 mL/g	20.69 ± 12.83	2.01 ± 0.39	7.45 ± 3.59	
	50 mL/g	6.68 ± 3.02	2.58 ± 1.51	14.21 ± 7.58	

Leaves (*t*–test for independent extracts): ^1^ difference in chlorogenic acid between 10 and 30 mL/g and 10 and 50 mL/g; ^2^ difference in rutin content between 10 and 50 mL/g; Waste (Kruskal–Wallis ANOVA test): ^3^ difference in chlorogenic acid; ^4^ difference in rutin content.

**Table 8 molecules-24-01611-t008:** Optimal UAE conditions for the extraction of chlorogenic acid and rutin.

Sample	Leaves	Scrap	Dust	Midrib
Optimal UAE parameters for chlorogenic acid	44.5 °C, 17.23 min, 11 mL/g, 40.46% ethanol-water ratio	46.69 °C, 15.19 min, 10 mL/g, 40% ethanol-water ratio	53.59 °C, 38.31 min, 10 mL/g, 55.43% ethanol-water ratio	30.14 °C, 38.31 min, 11 mL/g, 44.83% ethanol-water ratio
Predicted chlorogenic acid content (μg/mL)	276.734	78.49	49.44	12.24
Optimal UAE parameters for rutin	44.71 °C, 42.76 min, 49.73 mL/g, 76.83% ethanol water ratio	69.72 °C, 26.23 min, 49.8 mL/g, 59.12% ethanol water ratio	37.78 °C, 43.02 min, 49.88 mL/g, 58.43% ethanol water ratio	69.27 °C, 39.716 min, 49.06 mL/g, 74.44% ethanol water ratio
Predicted rutin content (μg/mL)	71.83	23.87	17.52	6.0185

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
