# Peer review of "Optimization of Ultrasound-Assisted Extraction of Some Bioactive Compounds from Tobacco Waste"

_molecules, 2019, doi:10.3390/molecules24081611_

Round 1
Reviewer 1 Report
The manuscript entitled "Optimization of ultrasound-assisted extraction of some bioactive compounds from tobacco waste" investigated the effect of ultrasound-assisted extraction (UAE) on bioactive compounds from different types of tobacco industry waste (scrap, dust and midrib). Although the experiments appear to be well planned, results seem interesting and correct, the ideas and methods are standard and the results are solid, however, the topic does not contain new ideas and the discussion section is standard and not surprising. In my mind, this article is not acceptable for publication in Molecules.Author Response
Thank you for your comments.
We agree that there are several paper dealing with ultrasound-assisted extraction of tobacco leaves and waste and we presented them in Table 2 but these papers either fail to consider different types of tobacco waste or consider tobacco waste generally. Our results clearly shows that there are significant differences between different types of tobacco waste and that they should consider as individual plant materials. Moreover, our study consider optimization of extraction processes and statistical analysis what make our results much stronger. To the best of our knowledge, there are still no published paper deals with ultrasound-assisted extraction of tobacco ways systematically, taking into account process condition, type of material, comparison with starting material, statistical analysis and optimization.
Reviewer 2 Report
Review for molecules-485338 and manuscript entitled: “Optimization of ultrasound-assisted extraction of some bioactive compounds from tobacco waste” In my opinion the manuscript can be accepted for publication after extensive improvements that is detailed below.
Detailed suggestions:
1. In Tables 2 and 3, please explain why different results were obtained in the same extraction conditions.
2. The order of data was confusion in Table 2, so that it was causing troubles when comparing with the content in "results and discussion" section. Please reorder the data.
3. In line 125, “Content of chlorogenic acid varied between 54.9 and 2720 μg/mL in leaves extracts,….” should be change to “Content of chlorogenic acid varied between 44.3 and 2720 μg/mL in leaves extracts,….”
4. In line 130, “Low content of caffeic acid in all extracts, including tobacco leaves (4.6-31.5 μg/mL)….” should be change to “Low content of caffeic acid in all extracts, including tobacco leaves (3.6-31.5 μg/mL)….”
5. In line 137, “Content of rutin in tobacco leaves varied between 4.22 μg/mL and 427 μg/mL similar to….” should be change to “Content of rutin in tobacco leaves varied between 77.6 μg/mL and 427 μg/mL similar to….”
6. In line 176~177, the manuscript showed the conclusion of “The antioxidant activity in obtained extracts followed the order leaves extract>scrap extract >dust extract >midrib extract.”, but there was no evidence to support this conclusion.

Author Response
Reviewer
Comments and Suggestions for Authors
Review for molecules-485338 and manuscript entitled: “Optimization of ultrasound-assisted extraction of some bioactive compounds from tobacco waste” In my opinion the manuscript can be accepted for publication after extensive improvements that is detailed below.
Detailed suggestions:
Comment: In Tables 2 and 3, please explain why different results were obtained in the same extraction conditions.
Answer: We appreciate this comment. It is true that at some same extraction conditions different results were obtained but those differences were not too big and didn’t affected on our results in statistical analysis and optimization. Probably they occurred as experimental error, including human error, systematic error or random error caused by environmental conditions or other unpredictable factors. That is the reason, why proper experimental design (Box-Behnken design in our case) is necessary to minimize experimental error and produce the most accurate data possible. Moreover, if you have almost the same data in central points (0,0,0) then lack of fit is significant and model is not adequate.
Comment: Line 45. The order of data was confusion in Table 2, so that it was causing troubles when comparing with the content in "results and discussion" section. Please reorder the data.
Answer: We appreciate this comment. Data are reordered.
Comment: In line 125, “Content of chlorogenic acid varied between 54.9 and 2720 μg/mL in leaves extracts,….” should be change to “Content of chlorogenic acid varied between 44.3 and 2720 μg/mL in leaves extracts,….”
Answer: Revised as suggested.
Comment: In line 130, “Low content of caffeic acid in all extracts, including tobacco leaves (4.6-31.5 μg/mL)….” should be change to “Low content of caffeic acid in all extracts, including tobacco leaves (3.6-31.5 μg/mL)
Answer: Revised as suggested
Comment: In line 137, “Content of rutin in tobacco leaves varied between 4.22 μg/mL and 427 μg/mL similar to….” should be change to “Content of rutin in tobacco leaves varied between 77.6 μg/mL and 427 μg/mL similar to….”
Answer: Revised as suggested.
Comment: In line 176~177, the manuscript showed the conclusion of “The antioxidant activity in obtained extracts followed the order leaves extract>scrap extract >dust extract >midrib extract.”, but there was no evidence to support this conclusion.
Answer: We appreciate this comment. It was mistake. The sentence is removed from the conclusion part and replaced with other sentence.
Reviewer 3 Report
The authors submitted a manuscript on the “Optimization of ultrasound-assisted extraction of some bioactive compounds from tobacco waste”.
The topic of this paper is of scientific importance and within the scope of “Molecules”.
The abstract provides a clear overview of the experimental work and of the most relevant results obtained. The keywords are appropriate.
In the introduction section, the objectives of the experimental work are introduced. A critical presentation of the state-of-the-art was provided. The authors aimed at evaluating the possibility to recover bioactive compounds from tobacco waste, in particular scrap, dust, midrib and leaves via ultrasound-assisted extraction (UAE).
An optimization of UAE process parameters was carried out employing a Box-Behnken experimental design coupled with response surface methodology. HPLC was employed to identify phenolics and solanesol. Furthermore, different extraction procedures were followed for the extraction of phenolics and solanesol.
The “materials and methods” section accurately describes the statistical, analytical and experimental methods employed.
The results are sorted in subparagraphs. The optimization process allowed individuating the optimal conditions to achieve the maximum antioxidant activity and concentration of targeted compounds.
Tables and figures are clear and self-explanatory. The results are supported by literature contributions and a critical discussion is provided.
Therefore, the conclusions are supported by the results and the objectives of the work were fulfilled.
Overall, English language is acceptable.
Author Response
Authors are grateful to the reviewer for his positive and encouraging comments.
Reviewer 4 Report
Interesting manuscript considering the quantity of by-products of tobacco processing, their chemical content and the possible isolation of products, one of all solenasol, difficult to obtain by synthesis and at the same time characterized by a wide range of properties and possible uses

Author Response
We are very thankful for this valuable comments. We have corrected al issues pointed out in the manuscript and tables.
Round 2
Reviewer 2 Report
Accept in present form